# Cell Surface Expression of Nrg1 Protein in *Candida auris*

**DOI:** 10.3390/jof7040262

**Published:** 2021-03-31

**Authors:** Anuja Paudyal, Govindsamy Vediyappan

**Affiliations:** Division of Biology, Kansas State University, Manhattan, KS 66506, USA; anuja@ksu.edu

**Keywords:** *Candida auris* biofilm, drug-resistant, sweat medium, cell surface proteins, BME extract, 2-DE, Zn-finger protein, cell surface-associated Nrg1, anti-Nrg1 antibody, immunofluorescence

## Abstract

*Candida auris* is an emerging antifungal resistant human fungal pathogen increasingly reported in healthcare facilities. It persists in hospital environments, and on skin surfaces, and can form biofilms readily. Here, we investigated the cell surface proteins from *C. auris* biofilms grown in a synthetic sweat medium mimicking human skin conditions. Cell surface proteins from both biofilm and planktonic control cells were extracted with a buffer containing β-mercaptoethanol and resolved by 2-D gel electrophoresis. Some of the differentially expressed proteins were excised and identified by mass spectrometry. *C. albicans* orthologs Spe3p, Tdh3p, Sod2p, Ywp1p, and Mdh1p were overexpressed in biofilm cells when compared to the planktonic cells of *C. auris*. Interestingly, several proteins with zinc ion binding activity were detected. Nrg1p is a zinc-binding transcription factor that negatively regulates hyphal growth in *C. albicans*. *C. auris* does not produce true hypha under standard in vitro growth conditions, and the role of Nrg1p in *C. auris* is currently unknown. Western blot analyses of cell surface and cytosolic proteins of *C. auris* against anti-*Cal*Nrg1 antibody revealed the Nrg1p in both locations. Cell surface localization of Nrg1p in *C. auris,* an unexpected finding, was further confirmed by immunofluorescence microscopy. Nrg1p expression is uniform across all four clades of *C. auris* and is dependent on growth conditions. Taken together, the data indicate that *C. auris* produces several unique proteins during its biofilm growth, which may assist in the skin-colonizing lifestyle of the fungus during its pathogenesis.

## 1. Introduction

Skin, a major interface between host and environment, is a habitat for several microbial populations. Many skin residing microbes have the ability to form skin biofilms that can lead to skin infection, and in certain conditions, can enter the body to cause life threatening bloodstream infections [1].

*Candida auris* is an emerging multi-drug resistant human fungal pathogen detected in many countries simultaneously [2]. It colonizes skin surfaces and can be invasive in immunocompromised patients [3]. *C. auris* has been isolated from blood, wounds, body fluids, and skin [4], where bloodstream infection is the most common infection leading to a mortality rate as high as 30–60% [5]. Additionally, the ability of *C. auris* to adhere to and to form drug-resistant biofilms has been a serious issue [6,7]. Until recently, *C. albicans* has been considered the major pathogen but recently, there has been increasing evidence showing *C. auris* involved in biofilm formation resulting in fatal bloodstream infections [8]. Unlike *C. albicans*, *C. auris* does not produce hyphae under in vitro conditions but it can form hyphae in vivo [9] or elongated pseudo hyphae in the presence of genotoxic compounds [10].

Nrg1p is a zinc-finger domain containing DNA-binding protein and is a key transcription factor that negatively regulates hyphal growth in *C. albicans* [11,12]. The *NRG1* transcript is down-regulated during hypha-inducing growth conditions, and its deletion has caused constitutive filamentous growth, similar to the *tup1* null mutant under yeast-promoting growth conditions because the expression of hypha-specific genes (e.g., *UME6*, *ECE1*, *HWP1*, etc.) was derepressed [12]. In a transcriptomic study, hundreds of genes were found to be differentially regulated in a *NRG1*-dependent and *NRG1-TUP1*-dependent manner. Studies have also shown that *C. albicans* Nrg1p plays important roles in biofilm formation and dispersion, and deletion or overexpression of *NRG1* blocks the yeast-to-hypha transition resulting in attenuated virulence [12,13,14]. However, the expression of *C. auris NRG1* and its role in virulence is unknown.

Sweat media that mimics axillary skin condition promotes *C. auris* biofilm formation in vitro and biofilm produced by *C. auris* in this media was ten times more robust than that of biofilms formed by *C. albicans*. According to a recent study, high salinity and fatty acids mimicking sweat favors growth of *C. auris* allowing it to survive for 14 days whereas *C. albicans* survived for less than a week [15]. Since cell surface proteins play a major role in host cell adherence and biofilm formation, understanding their expression and their cellular roles in the pathogenicity of *C. auris* will help develop effective intervention strategies. We used sweat medium to study *C. auris* biofilm proteins.

In this study, we extracted non-glucan attached cell surface proteins from *C. auris* biofilm and planktonic cells, separated by 2-d gel electrophoresis, and identified some of the differentially expressed proteins by liquid chromatography coupled with mass spectrometry. We have also demonstrated an unexpected finding of cell surface localization of *C. auris* Nrg1p, which is a zinc-finger transcription factor and is predicted to be nuclear.

## 2. Materials and Methods

### 2.1. C. auris Strains and Growth Conditions

*Candida* species were routinely maintained on YPD (1% yeast extract, 2% peptone, and 2% dextrose) agar (1.5%). *Candida auris* isolated from Saudi Arabia by Abdalhamid et al. [16] (South Asian, clade-I) was used for most of the experiments in this study. A single colony of this *C. auris* was inoculated into YPD broth medium and grown overnight at 37 °C with shaking at 200 rpm. For planktonic cells, overnight grown cells were used to inoculate 100 mL of fresh sweat medium to make a suspension of approximately 2 × 10^5^ cells at an optical density of 0.05 (OD_600_) and grown at 37 °C for 48 h with shaking (200 rpm). Sweat medium was prepared as mentioned by Horton et al. [15]. Biofilms were formed in cell culture treated dishes using the same cell density as for planktonic growth. They were allowed to grow at 37 °C for 48 h without disturbance. For pseudohyphal induction by hydroxyurea (HU) in *C. auris* [10], the sweat medium was used with and without 100 mM HU and grown statically for 48 h at 37 °C. *C. auris* strains (Table 1) (Clades II-IV), representing those from different countries, were obtained from the Centers for Disease Control and Prevention (CDC) and the Food and Drug Administration (FDA) Antibiotic Resistance (AR) isolate bank (Atlanta, GA, USA).

### 2.2. Cell Surface Proteins Preparation

Planktonic cells were collected by centrifugation while biofilm cells were scraped from cell culture dishes using a cell scraper. Fungal cells were washed twice with sterile phosphate buffered saline (PBS), pH 7.5, to remove residual media components. Cell surface proteins (non-glucan bound) were extracted from both planktonic and biofilm cells by resuspending them in ammonium carbonate (1.89 g/L) buffer containing 1% β-mercaptoethanol (BME) and incubated at 37 °C for 30 min as described [19,20]. Pooled BME extracts were dialyzed thoroughly against water and lyophilized before analyses.

### 2.3. 2-Dimensional Gel Electrophoresis (2-DE)

Lyophilized cell surface proteins were resuspended in a small volume of sterile water and their concentration determined using the Bradford dye-binding method [21]. 2-DE was performed using the immobilized pH gradient (IPG) gels (Immobiline Drystrip, pH 3–10, NL, 7 cm, Cytiva, MA, USA). Aliquots of protein suspensions were diluted in IPG buffer as described by the manufacturer and applied to IPG gels. The gel strips were rehydrated overnight at room temperature. First dimension electrofocusing was performed using Multiphore-II unit (Pharmacia Biotech, Uppsala, Sweden) at 20 °C, 3500 V for 6 h. Gels were equilibrated first in an equilibration buffer containing dithiothreitol for 15 min, then in a buffer containing iodoacetamide for another 15 min as recommended by the manufacturer. Electrofocused gel strips were run in the second dimension using 12.5% SDS-PAGE for 90 min at 100 V. Resolved proteins were stained by silver stain (Sigma-Aldrich, St. Louis, MO, USA) and images were recorded against a white light background. Cell surface proteins from at least two different biological samples were analyzed by 2-DE. Proteins that were differentially expressed were identified and some were excised for LC-MS/MS analysis. When necessary, 2-DE separated proteins were transferred to an Immobilon-P polyvinylidene fluoride (PVDF) membrane (Millipore Sigma, Billerica, MA, USA) electrophoretically for Western analysis.

### 2.4. Mass Spectrometry Analysis

Proteins from 2-DE were analyzed by a core facility service (DNA/Protein Resource Facility, Oklahoma State University, Stillwater, OK, USA). Proteins from the excised gel spots were digested with trypsin and collected. Peptides were dissolved in 0.1% formic acid, and injected onto a 75-micron × 55 cm nano HPLC column packed with 2-micron C18 media (Thermo Fisher PN 164942). Peptides were separated using 0.1% aqueous formic acid as mobile phase A and acetonitrile/water/formic acid (80:20:0.1) as mobile phase B to develop a gradient of 0–35% over 60 min. The column was terminated with stainless-steel emitter within a Nanospray Flex Ion source (Thermo Fisher, Waltham, MA, USA). The ion stream was analyzed in a quadrupole-Orbitrap tribrid mass spectrometer (Orbitrap Fusion model, Thermo Fisher, Waltham, MA, USA), using a data-dependent 5-s Top Speed method, wherein peptide precursors were measured in the Orbitrap sector, isolated in the quadrupole sector and fragmented in the ion routing multipole. After dissociation, peptide fragment ions were analyzed in the ion trap sector. The results were analyzed using MaxQuant software and the *C. auris* B8441 reference genome. Criteria such as peptide ion signal peak intensity, peptide spectral match (PSM)-based quantity, *Q* values, peptide mass, pI, and sequence coverage were included during analysis.

### 2.5. Bioinformatic Analyses

The amino acid sequence of Nrg1p from *C. albicans* (C7_04230W_A, Candida Genome Database, CGD) was blasted (NCBI blast) against that of *C. auris* (B9J08_005429, B8441, CGD) to determine the percent homology between them. Iterative Threading ASSEmbly Refinement (I-TASSER) [22] was used to generate a 3D model of *C. auris* and *C. albicans* Nrg1 protein based on previously available 3D structures of one or more closely related proteins. Using 3D models from I-TASSER, structural alignment of *C. auris* and *C. albicans* Nrg1 was carried out using RaptorX structural alignment tool [23].

### 2.6. NRG1 Cloning, Nrg1p Purification and Antibody Production

The gene (*NRG1*) encoding Nrg1p of *C. albicans* was synthesized and cloned in-frame in an *Escherichia coli* overexpression vector (pET24a, Kan^R^, *Nde*I and *Xho*I, Epoch Life Sciences Inc., Missouri City, TX, USA) that contains a 6-Histidine tag at its C-terminus. The plasmid was transformed into *E. coli* (BL21-DE3 or Rosetta) cells. Nrg1p was overexpressed by induction with 1 mM of isopropyl-β-d-1-thiogalactopyranoside (IPTG) in antibiotic containing medium. Expressed cells were pelleted and resuspended in phosphate buffer containing a protease inhibitors cocktail followed by cell lysis using a French press (19,000 psi) with centrifugation at 10,000× *g* for 10 min at 4 °C. Nrg1p from the supernatant was purified using nickel-nitrilotriacetic acid (Ni-NTA) coupled agarose beads (Qiagen, Germantown, MD, USA) as recommended by the manufacturer. Thus, the purified Nrg1p was dialyzed against water, verified for homogeneity on SDS-PAGE and used for antibody production (*Cal*Nrg1 antibody) in a rabbit (LAMPIRE Biological Laboratories, Pipersville, PA, USA).

### 2.7. Immunoblotting

*Cal*Nrg1 polyclonal antibody was used to detect *C. auris* Nrg1p. Cell surface and cytosolic proteins from both planktonic and biofilm cells were separated by SDS-PAGE after boiling and reducing conditions and transferred to an Immobilon-P PVDF membrane (Millipore Sigma, Billerica, MA, USA). The membrane was blocked using 5% non-fat dry milk in a buffer containing Tris-HCl, NaCl, and Tween-20 and blotted against an anti-rabbit *Cal*Nrg1 antibody followed by an anti-rabbit IgG-HRP conjugate (R & D System, Minneapolis, MN, USA). The ECL Western Blotting Substrate (Thermo Scientific, Waltham, MA, USA) was used to detect the protein bands followed by imaging in Azure c600.

### 2.8. Immunofluorescence

*C. auris* cells grown in sweat medium for 48 h at 37 °C were collected and washed twice using PBS to remove media constituents. Immunofluorescence of *C. auris* after immunostaining was performed as described [24] with slight modifications. Briefly, cells were fixed in 4% formaldehyde for 15 min followed by a PBS wash. Fixed cells were then blocked with 3% BSA in PBS containing 0.1% Tween-20, washed with PBS and incubated with 1:100 anti-*Cal*Nrg1 antibody raised in a rabbit overnight at 4 °C. Cells were washed twice with PBS and further incubated in an anti-rabbit Alexa Fluor-secondary antibody (Thermo Fisher Scientific, Waltham, MA, USA) 1:100 for 2 h. Cells were washed three times and mounted on a slide to examine under a fluorescence microscope (Leica DM 6 B with a camera attached).

## 3. Results

### 3.1. Cell Surface Proteins Are Differentially Expressed in Planktonic and Biofilm Cells

Cell surface proteins play a major role in adhesion and biofilm formation. To identify the differentially expressed non-glucan attached cell surface proteins from biofilm and planktonic *C. auris* cells (strain #1126, Clade I, Table 1), we used proteomics approaches. Using 2-D gel electrophoresis followed by silver staining, several differentially expressed proteins can be found and representative images are shown (Figure 1a,b). Selected protein spots from the biofilm sample were excised from the gel and analyzed by mass spectrometry. These cell surface proteins are listed in Table 2. *C. albicans* ortholog proteins including Spe3p, Tdh3p, Sod2p, Ywp1p and Mdh1p were present in high intensities in *C. auris* biofilm cells when compared to planktonic cells.

Protein spot #1 is spermidine synthase (Spe3p). It localizes in the extracellular region and plays role in spermidine and pantothenate biosynthesis. Similarly, Tdh3p was also expressed more in the biofilm cells which are known to be located in the cytoplasm, cell wall and cell surface regions. An increased *TDH3* transcript has been reported in *C. albicans* biofilm [25]. The next protein that showed increased expression in biofilm cells was Sod2p (spot #3) which has superoxide dismutase activity and is known to be found in the mitochondrial matrix [26,27]. Ywp1p is a yeast wall protein-1 found on the cell surface and cell wall of yeasts known to be involved in adhesion and biofilm formation [28]. Mdh1 is a predicted malate dehydrogenase precursor and is found in the extracellular regions and mitochondria. It was upregulated in *C. albicans* when grown as a biofilm [29]. Future studies are required to identify additional proteins that are differentially expressed and validate them.

### 3.2. Nrg1 Protein Expresses in C. auris Biofilm and Planktonic Cells

*C. albicans* is known to cause biofilm-related infections and strains that are defective in hyphal growth fail to produce robust biofilm and matrix polysaccharides [30,31]. Nrg1p is a negative regulator of hyphal growth in *C. albicans,* but its role and expression of Nrg1 in *C. auris* is unknown. Since *C. auris* produces yeast cells during in vitro biofilm growth and a gene encoding Nrg1p is present in *C. auris* (B8441 genome, B9J08_005429, *C. albicans* ortholog), we were interested in investigating it in *C. auris*. Further, Nrg1p is a zinc-binding transcription factor and we identified other zinc-binding proteins including Fba1p, Adh1p, Xyl2p and Pmi1p in our MS analysis, prompting us to investigate Nrg1p further.

To determine Nrg1p expression and localization in *C. auris*, we used an anti-*Cal*Nrg1 antibody (polyclonal) that we raised in our laboratory. Briefly, *C. albicans NRG1* with 6His tag at the C-terminus was cloned, overexpressed and purified in an *E. coli* expression system. The purified Nrg1 protein with a molecular weight of approximately 36 kDa reacted to both the anti-His and anti-*Cal*Nrg1 antibody (Figure 2a). To examine Nrg1 protein expression in *C. auris*, it was grown in sweat medium as planktonic and biofilm cells. Using the *Cal*Nrg1 antibody, we detected an Nrg1 protein band of ~52 kDa in both the *C. auris* cell surface and cytosolic protein extracts of biofilm and planktonic cells (Figure 2b). The predicted molecular weight of *C. auris* Nrg1p is 26.1 kDa (B9J08_005429) and the 52 kDa reactive band is likely to be a dimer of Nrg1p. Nrg1p is expressed slightly more in the cell surface proteins of biofilm than that of planktonic cells (Figure 2b) even though an equal amount of protein was used (Ponceau staining, data not shown). The antibody reacted to *C. albicans* cytosolic Nrg1 protein but none in the cell surface proteins (data not shown). The *Cal*Nrg1 antibody did not react to the cytosolic proteins of *C. albicans nrg1*^−/−^ (null) mutant [18], consistent with its specificity for Nrg1p (Figure 2b). Anti-GAPDH (glyceraldehyde 3-phosphate dehydrogenase)antibody (human, horseradish peroxidase (HRP)-conjugated, Santa Cruz Biotechnology, TX, USA) reacted with a cytosol protein component but not with a cell surface protein component.

### 3.3. Nrg1 Proteins of C. albicans and C. auris Are Structurally Similar

As a transcription factor, Nrg1p is predicted to be in the nucleus in *C. albicans* and the cell surface expression of this protein is unexpected. To understand the similarities of Nrg1 proteins between *C. auris* and *C. albicans*, we compared their amino acid sequences and aligned their predicted 3D structures using software. *C. albicans* and *C. auris* Nrg1 proteins contain 310 and 236 amino acids with predicted molecular weights of 34.3 kDa and 26.1 kDa, respectively. Sequence alignment using Clustal Omega shows a total of 38% identity (Figure 3a), and Basic Local Alignment Search Tool (BLAST) for protein result shows 67% identity in the region between 185 to 220 of *C. auris* and 219 to 254 region of *C. albicans* Nrg1p. Based on structure prediction software, five structures were predicted for both *C. auris* and *C. albicans* Nrg1. Structures with the least C-score were picked for both proteins (Figure 3b,c). Using these predicted structures, structural alignment was performed using RaptorX software (Figure 3d) and a template modeling (TM) score of 0.667 was obtained which suggests that these two proteins share a similar fold. Interestingly, the C-terminal C_2_H_2_ zinc-finger domains between these two proteins are highly conserved (Figure 3a–d). The zinc-finger domain is involved in DNA binding activity and this type of Class 1 zinc-finger protein is common in most of the eukaryotes [32].

### 3.4. Nrg1 Protein Localizes on C. auris Cell Surface

Having verified bioinformatically the similarities of Nrg1 proteins between *C. auris* and *C. albicans* and detected the Nrg1p in the cell surface extract of *C. auris* during the 1st-dimensional SDS-PAGE/Western analysis (Figure 2 and Figure 3), we next wanted to determine if Nrg1p can be detected from the cell surface fraction of *C. auris* after 2-DE. This is also because we did not identify Nrg1p in our initial proteomic analyses of *C. auris* biofilm cell surface proteins (2-DE and MS, Figure 1 and Table 2).

We used cell surface proteins (BME extract) collected from *C. auris* biofilm cells grown in the sweat medium and resolved by 2-DE as above and analyzed by Western blot against the *Cal*Nrg1 antibody (Figure 4b). The result shows a single protein spot that reacted to the *Cal*Nrg1 antibody that matches the predicted size of 26 kDa of *C. auris* Nrg1 protein (Figure 4b). To further validate the specificity of the *Cal*Nrg1 antibody to Nrg1 proteins of Candida species, the *Cal*Nrg1 antibody was depleted against the purified *C. albicans* rNrg1-6His protein and used for Western analysis. The depleted antibody failed to detect Nrg1 proteins of *C. auris* cytosolic proteins, suggesting its specificity to Nrg1p (Appendix A).

To determine the surface localization of the Nrg1p on *C. auris* cells *in vivo*, we performed an immunofluorescence assay. *C. auris* cells were grown in synthetic sweat medium for 48 h and used *Cal*Nrg1 antibody (primary antibody) followed by a fluorescent secondary antibody as described in the Materials and Methods Section.

Interestingly, we observed Nrg1p expression on the surface of *C. auris* cells where several punctate fluorescence signals can be seen (Figure 5). These fluorescence signals were discrete and found throughout the cell surface. Cells stained with a second antibody coupled with Alexa Fluor only (minus the primary antibody) did not show any fluorescent signal (Figure 5, lower panel).

### 3.5. Nrg1 Protein Expresses in Different Clades of C. auris and Is Dependent on Growth Conditions

To know if Nrg1p expresses in different clades of *C. auris*, we obtained a panel of *C. auris* strains (Clades II-IV) from the CDC and used one strain from each clade along with *C. auris* strain #1126 (Clade I) to analyze Nrg1p expression. Cells grown in the sweat medium were homogenized and the cytosolic proteins were used for Western analysis. Results shown in Figure 6a indicate Nrg1p expresses in all the four strains representing different clades. We also determined the expression of Nrg1p in a rich medium, YPD broth, and found it was poorly expressed in this medium when compared to the sweat medium (Figure 6b).

### 3.6. Yeast-Hypha Regulation in C. auris Could Be Different from C. albicans

*C. albicans* produces hyphae under hypha-inducing conditions (growth temperature at 37 °C, RPMI-1640 medium and media containing serum, etc.) whereas *C. auris* remains in the yeast form under these conditions, hinting that its hyphal regulatory pathway(s) may be non-functional or different. However, *C. auris* forms elongated pseudohyphal phenotype when treated with a genotoxic compound, hydroxyurea (HU) [10], while the control cells (without HU) were mostly yeast cells which allowed us to determine whether *C. auris* Nrg1 plays any role in the pseudohyphal growth. Western analysis of cytosolic proteins from these two growth forms showed no differences in the expression levels of Nrg1p (Figure 7).

## 4. Discussion

*Candida auris*, a newly emerged drug-resistant human fungal pathogen, was first reported in 2009 in Japan [33] and since then it has been isolated in over 40 countries across 6 continents [34]. *C. auris* is resistant to almost all clinical antifungal drugs and the CDC has listed this pathogen at an urgent threat level (fact sheet, CDC).

Although *C. auris* can be isolated from multiple sites in the body, it predominantly colonizes the skin and also persists in the environment, particularly in healthcare settings. Skin infections caused by *C. auris* in humans could be in a biofilm status. Despite the fact that *C. auris* produces fewer dense biofilms than *C. albicans* [35,36], *C. auris* can survive at high salt and other stress conditions [37] normally found in the skin niche and able to produce biofilms. Researchers have shown that *C. auris* is able to form 10-fold more biofilm than that of *C. albicans* in a skin-mimicking synthetic sweat medium that contains high salts and various lipids [15]. *C. auris* also has a high ability to colonize the skin, as demonstrated using porcine skin as an ex vivo model [15]. The high tolerance of this fungus to salty sweat medium mimicking skin conditions may play a role in its pathogenicity and persistence. We, therefore, as a first step, aimed to analyze the cell surface proteins from *C. auris* biofilms grown in the sweat medium.

Fungal cell surface proteins play a major role in pathogenesis. These proteins are one of the major groups of molecules that mediate the initial contact with its surrounding environment. Some cell surface proteins are covalently attached to mannan and other polysaccharides. Earlier studies have shown that the cell surface proteins can be extracted using an alkaline buffer containing β-mercaptoethanol (BME) [19,20,38]. By employing this method, we extracted a covalently attached cell surface and other soluble proteins from *C. auris* biofilms developed in sweat medium. As a control, we used planktonic cells grown in the same medium. Using 2-DE and silver staining, several differentially expressed proteins were detected from the BME extract of biofilm cells (Figure 1). This result shows that the *C. auris* biofilm produces some cell surface proteins at a higher level compared to planktonic cells. Although a limited number of protein spots were excised and analyzed by mass spectrometry, several of those detected are known to be found in the cell wall and/or on the cell surface. 

Upregulated proteins that we identified by proteomics of biofilm cells include Spe3, Mdh1, Sod2, Ywp1 and Tdh3 (Table 2). Spe3p is involved in the biosynthesis of spermidine in *C. albicans* and also in many other fungi [39,40]. Spermidine is one of the three types of polyamines (putrescine and spermine are the others) that play a major role in various cellular processes including growth, oxidative and osmotic stress responses [39]. It is worth noting that *C. auris hog1* mutant was acutely sensitive to organic oxidative- and osmotic-stress-inducing agents [37]. Similarly, increased expression of Sod2p, the mitochondrial Mn-containing superoxide dismutase in *C. auris* biofilms, may provide antioxidant properties to the fungus against environmental or host oxidative stress. These proteins were shown to be overexpressed in *C. albicans* biofilms [41]. Certain proteins are involved in more than one cellular functions as in the case of *C. glabrata* Tdh3p that expressed more in biofilm than planktonic cells [42]. Proteomic analyses of cell surface proteins (BME) and extracellular vesicles of *C. albicans* yeast and hyphal morphologies were shown to contain Tdh3p [43,44]. Since Tdh3p is a moonlighting protein (involved in glycolysis, oxidoreductase activity and host–pathogen interaction), it may have some of these roles in *C. auris* biofilm growth. Future genetic studies (gene deletion or controlled expression) will determine the functional roles of these proteins in *C. auris* biofilm growth and virulence. *C. albicans* is known to cause biofilm-related infections and strains that are defective in hyphal growths fail to produce robust biofilms [30,31]. In contrast to the *C. albicans* biofilm production phenotype, *C. auris* produces ten-fold more biofilm without producing hyphae but with yeast cells [15]. This distinctive *C. auris* phenotype prompted us to search for a major negative regulator(s) of hyphal growth in *C. auris*. In conjunction with this notion, we have also identified some zinc-binding proteins in our 2-DE mass spectometry analyses. Nrg1p is a zinc-binding transcription factor that regulates hyphal growth negatively in co-operation with Tup1p in *C. albicans* [11], and the role and expression levels of Nrg1p in *C. auris* is unknown, although an ortholog is present. Further, Tup1, a zinc transcription factor similar to Nrg1p, has no hyphal regulatory function in *C. auris* [10].

Using an anti-Nrg1 antibody, we showed that Nrg1p is present in the cell surface extracts from both types of cells (Figure 2b). To confirm that the cell surface proteins are not contaminated with the cytosolic proteins during BME extraction, we probed these proteins with an anti-GAPDH (human) antibody that reacts only to the cytosolic proteins but not to the cell surface proteins (Figure 2b, lower panel). The presence of Nrg1p in the BME extract of *C. auris* biofilm cells was further demonstrated by a 2-DE resolved Western blot (Figure 4b). Interestingly, the immunofluorescence assay clearly showed the presence of Nrg1p on the cell surfaces of *C. auris* in vivo (Figure 5). Taken together, a zinc transcription factor Nrg1p, normally found in the nucleus, can also be present on the cell surface of *C. auris* cells. However, the mechanism of Nrg1p transport to the cell surface and its function(s) on the cell surface remain to be determined. 

Structural data for fungal Nrg1p are not available. However, the three-dimensional prediction of Nrg1 proteins from *C. auris* and *C. albicans* by the I-TASSER program and their structural alignment by RaptorX programs rendered a superimposed model showing Nrg1 proteins from both species have a similar fold (Figure 3b–d). Importantly, the zinc-binding and the DNA-binding C-terminal regions are highly conserved and similar in structure, including the alpha-helices and ß-sheet folds of the DNA binding regions of the Class-I zinc-finger transcription factors (Figure 3b–d) [32]. It is unclear if the lack of 44 residues at the C-terminus of Nrg1 in *C. auris* may have functional implications.

It is reasonable to speculate that the hyphal regulatory role of Nrg1p in *C. auris* may be different or absent from *C. albicans* Nrg1p. For example, *C. albicans* hyphae and pseudohyphae were shown to convert into yeast cells when they were treated with hyphal growth inhibitors called gymnemic acids (GAs) [45]. However, when pseudohyphae of *C. auris* were treated with GAs, they did not revert to yeasts and they remain pseudohyphae. Further, the expression level of Nrg1p in yeast and pseudohyphal growth forms of *C. auris* without and with HU, respectively, did not show any differences between the two growth forms (Figure 7) which may suggest that Nrg1p may have a limited role in the yeast-hyphal morphogenesis in *C. auris*. Alternatively, the growth condition or host niche-dependent cues may regulate hyphal morphogenesis in this fungus. When *C. auris* was grown in YPD with 10% NaCl, elongated pseudohyphal-like cells were formed [46]. Similarly, *C. auris* can filament when it is passaged in the mammalian body [9]. It is also worth mentioning that the representative strains from all four clades (Clade I-IV) that we analyzed contain Nrg1p (Figure 6a). Thus, *C. auris* is more tolerant to various stress conditions when compared to other *Candida* species which might help in the unique lifestyle of *C. auris*.

In conclusion, *C. auris* produces several differentially expressed cell surface proteins during its biofilm growth. Despite its predominant yeast form of growth, *C. auris* produces robust biofilm, and the Nrg1 ortholog is expressed on the cell surface of *C. auris*. While the function of Nrg1p is currently unknown, representative strains from all four known clades of *C. auris* express Nrg1p.

## Figures and Tables

**Figure 1 jof-07-00262-f001:**
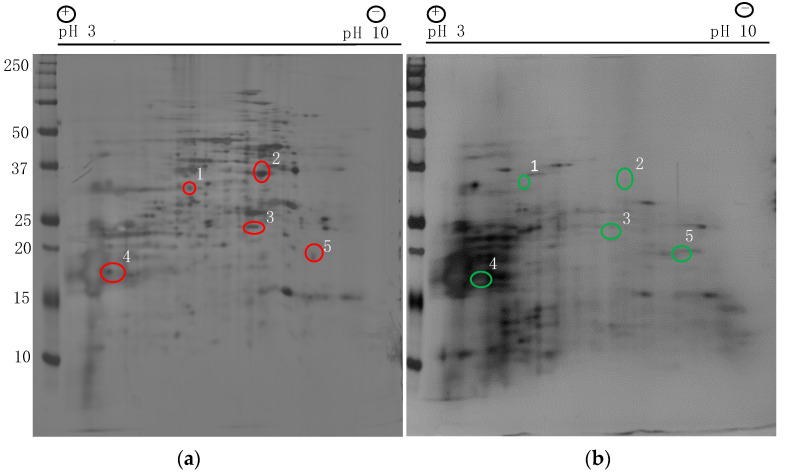
Cell surface proteins extracted from *Candida auris* biofilm and planktonic cells and resolved by 2-Dimensional Gel Electrophoresis (2-DE). Silver stained gel images of proteins from biofilm (**a**) and planktonic cells (**b**). Protein spots marked with red circles (**a**) are excised out and identified by mass spectrometry. These proteins were expressed less or were absent in planktonic cells (marked in green circles) (**b**).

**Figure 2 jof-07-00262-f002:**
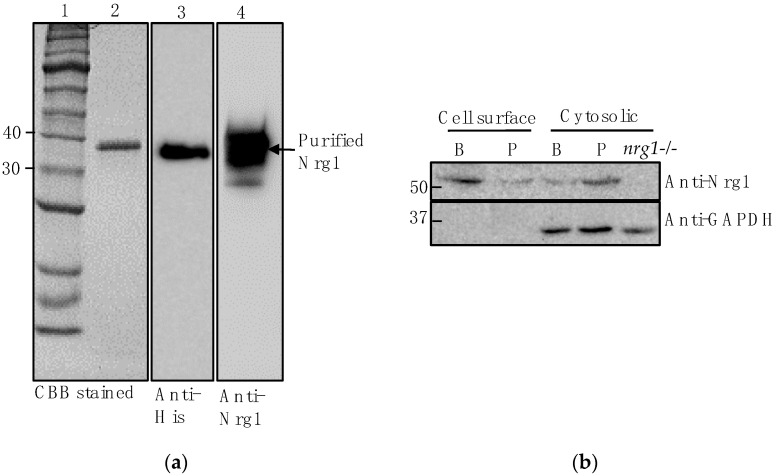
Purification of *C. albicans* Nrg1 protein overexpressed in *Escherichia coli* and the expression of Nrg1p in *C. auris* biofilm and planktonic cells. (**a**) Purified r*Cal*Nrg1-6His protein separated on SDS-PAGE. Lane 1. Protein ladder (New England Biolab, Boston, MA, USA), Lane 2. Ni^++^ affinity purified *Cal*Nrg1p (Coomassie blue stained), Lane 3. Western blot of purified *Cal*Nrg1p against anti-His antibody, and Lane 4. Western blot of purified *Cal*Nrg1p against anti-*Cal*Nrg1 antibody. (**b**) Western blot of cell surface and cytosolic proteins of *C. auris* biofilm (B) and planktonic (P) cells probed with anti-*Cal*Nrg1 antibody. *C. albicans nrg*^−/−^ was included as negative control. Anti-GAPDH HRP-conjugate provides loading and subcellular location controls.

**Figure 3 jof-07-00262-f003:**
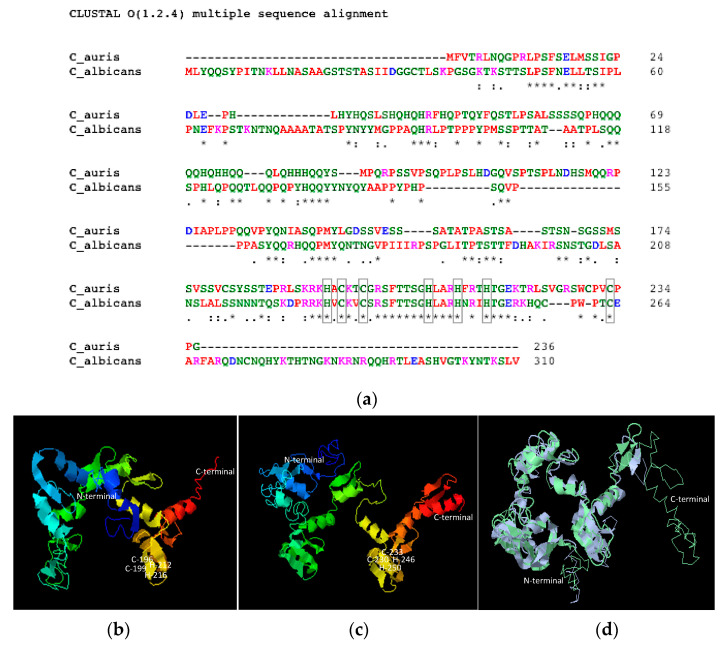
Comparison between Nrg1p of *C. auris* and *C. albicans*. Alignment of amino acid sequences of Nrg1p of *C. auris* and *C. albicans* (**a**). The Cys and His residues (C_2_H_2_ zinc-finger) and the identities between the amino acid residues are marked by gray boxes and asterisks, respectively. Predicted 3D structures of Nrg1p of *C. auris* (**b**) and *C. albicans* (**c**) using Iterative Threading ASSEmbly Refinement (I-TASSER). N-terminus, C-terminus and predicted zinc-binding domains are marked. (**d**) Structure alignment of *C. auris* Nrg1p (purple) and *C. albicans* Nrg1p (green) using predicted structures from I-TASSER. Note *C. auris* lacks 44 residues after the C_2_H_2_ zinc-finger domain (Class 1) at its C-terminus. The aligned image was tilted forward for better viewing, and hence the N-terminal regions appear different from images (**b**,**c**).

**Figure 4 jof-07-00262-f004:**
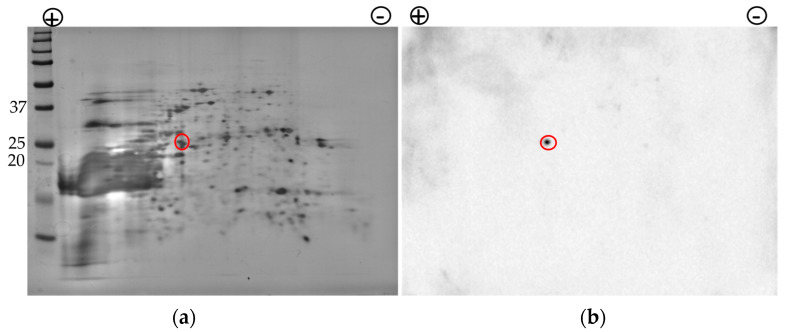
Nrg1p is localized in the cell surface extract of *C. auris*. (**a**), The silver-stained SDS-PAGE gel image of cell surface proteins after 2-DE. (**b**), Western analysis of the same proteins against rabbit anti-*Cal*Nrg1 antibody. The antibody reactive Nrg1p spot (marked in a red circle) with a molecular weight of around 26 kDa is shown in panel (**b**). All blue prestained protein standard (Bio-Rad, Hercules, CA, USA) was included in the 2nd dimension gel (**a**). Potential Nrg1p is marked in red circle (**a**).

**Figure 5 jof-07-00262-f005:**
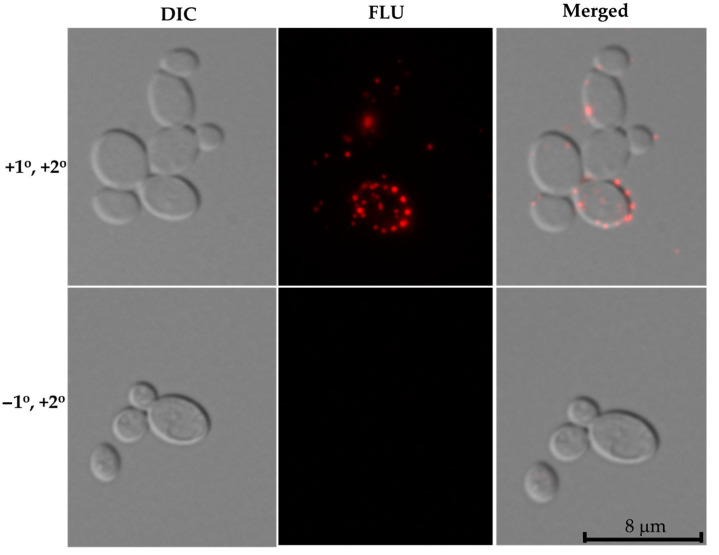
Immunofluorescence of cell surface expressed Nrg1p in *C. auris*. Synthetic sweat media grown *C. auris* cells were stained with anti-*Cal*Nrg1 antibody followed by anti-rabbit Alexa Fluor-secondary antibody and observed under a fluorescence microscope (upper panel). Cells without primary antibody (anti-*Cal*Nrg1 antibody) and with anti-rabbit Alexa Fluor-secondary antibody are showed in lower panel (control). Scale bar = 8 μm.

**Figure 6 jof-07-00262-f006:**
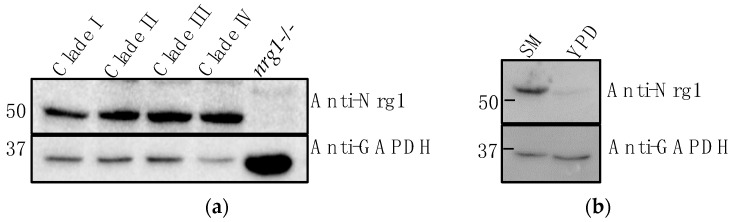
Nrg1p expression among different clades of *C. auris* and in different growth conditions. (**a**) Western blot of cytosolic proteins of 4 different clades of *C. auris*. Clade 1 (South Asia, Saudi Arabia, lab strain #1126), Clade II (AR bank #0381, East Asia), Clade III (AR bank #0383, Africa), Clade IV (AR bank #0386, South America). *C. albicans nrg1*^−/−^ null mutant was included as a negative control. Anti-GAPDH (human, HRP-conjugate) was used as a loading control. (**b**) Cytosolic proteins from *C. auris* grown in synthetic sweat medium (SM) and YPD broth medium were probed against anti-*Cal*Nrg1 antibody (upper panel). Equal amounts of protein loading were confirmed by use of anti-GAPDH-HRP conjugate (human) antibody.

**Figure 7 jof-07-00262-f007:**
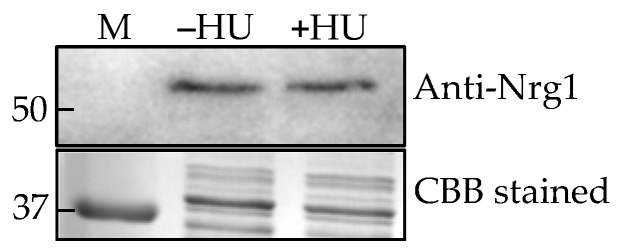
Nrg1p expression in *C. auris* treated with hydroxyurea (HU). Western analysis of cytosolic proteins from *C. auris* against *Cal*Nrg1 antibody (**upper panel**). Lane 1. Marker, Lane 2. Without HU, and Lane 3. With HU. SDS-PAGE gel of proteins from *C. auris* cells grown in sweat medium with and without HU followed by Coomassie brilliant blue (CBB) staining (**lower panel**).

**Table 1 jof-07-00262-t001:** List of Candida fungal strains used in this study.

Strains	Source
*Candida**auris* (Lab strain #1126, Clade I)	Abdalhamid et. al. (2018) [16]
*C. auris* (AR#0381, Clade II)	Antibiotic Resistance (AR) Isolate Bank, the Centers for Disease Control and Prevention (CDC) and the Food and Drug Administration (FDA)
*C. auris* (AR#0383, Clade III)	AR Isolate Bank, CDC/FDA
*C. auris* (AR#0386, Clade IV)	AR Isolate Bank, CDC/FDA
*C. albicans* (Lab strain #1)	Wildtype (SC5314) [17]
*C. albicans nrg1* ^−/−^	Noble’s deletion mutants [18]

**Table 2 jof-07-00262-t002:** List of proteins identified by MS. Proteins are selected on the basis of their peptide ion signal intensity, matching peptide and sequence coverage. *Q*-value of zero applies to all identified proteins.

Spot no.	Protein *C. auris*	Intensity	Number of Peptide	Seq Coverage (%)	Molecular Weight (kDa)		pH		Localization	Function	*C. albicans* Ortholog
					Predicted	Experimental	Predicted	Experimental			
1	A0A2H0ZC50	85,544,000	8	18.3	33.618	25–37	5.12	5.9	Extracellular region	Spermidine synthase	Spe3p
2	A0A2H1A5V9	131,870,000	20	66.2	34.541	37	6.81	6.5	Extracellular region, mitochondria	Malate dehydrogenase precursor	Mdh1p
3	A0A2H1A3N7	8,158,000	10	42.2	25.404	25	7.16	6.5	Cytoplasm, mitochondria	Superoxide dismutase	Sod2p
4	A0A2H0ZMT8	305,020	3	7	46.13	15–20	4.65	5	Cell surface, cell wall	Yeast form well protein	Ywp1p
5	A0A2H0ZM53	46,601,000	9	29.3	35.336	20	7.13	7.3	Cytoplasm, cell wall, cell surface	Glyceraldehyde 3-phosphate dehydrogenase activity	Tdh3

## Data Availability

Not applicable. (No gene expression or genomic data are involved).

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
