# Peer review of "Cell Surface Expression of Nrg1 Protein in Candida auris"

_jof, 2021, doi:10.3390/jof7040262_

Round 1

Reviewer 1 Report

The presented manuscript of A. Paudyal and G. Vediyappan about the cell surface protein expression in C.auris is interesting to read and deals with an important topic for the better understanding which factors may play a role in C.auris biofilm formation. Generally they focus on two very different strategies. First (1) they try to analyze the overall cell surface protein expression of planctonic and biofilm cells. Second (2) they focus on Nrg1p expression which seems a known relevant factor (in C.albicans) in biofilm formation. In the introduction nearly any information about Nrg1p is missing. The significance of Nrg1p in biofilm formation has to be described and documented by references. In the results section under the concept of (1) you have found 5 proteins highly expressed in biofilm C.auris cells. The presented experimental data are convincing. But the importance of these 5 proteins in C.auris biofilm formation remains unclear for the reader. Maybe the authors can carry out or at least propose experiments that may validate the real significance of these 5 proteins in C.auris biofilm formation. The (2) part dealing with Nrg1p is dominant in the manuscript and should be reflected in the title of the manuscript. Or this part (2) together with the yeast-hypha regulation part should be separated from the part (1) which seems a little bit uncomplete. The discussion of the findings under concept (1) - the five overexpressed proteins - is also nearly inexistant. In Figure 1 notation of (a) and (b) are missing.

Reviewer 2 Report

The authors of jof-1133032 manuscript aimed to study cell surface proteins from Candida auris biofilms grown in synthetic sweat medium. However, the paper mostly deals with one protein - Nrg1p, so the title of the manuscript should be adjusted to reflect that. The remaining comments are listed below.

1. Introduction

- The second paragraph has to be rewritten. In the current state, the presentation of information is chaotic and the meaning of sentences is often confusing. The problems with biofilm formation and persistence on surfaces have to be introduced in more detail.

- Lines 35-37: C. auris has been detected in many countries, not several ones.

- Lines 37-38: confusing sentence as these numbers apply to bloodstream infections only + reference missing.

- Lines 8-9 and 38-39: exaggeration, not all C. auris isolates are pan-resistant. to antifungals.

2. Materials and methods

- How many strains were used in the study? Were they all from the CDC/FDA AR Bank? -> Provide a list of strains used in the study.

- Not sure how references #10 and #11 are relevant for the "Single colony of Candida auris (South Asian, clade I)".

3. Results

- 3.1. Not clear which C. auris strain was used in 2-DE.

- Not clear why did the authors decide to investigate Nrg1p since it was not identified in 2-DE experiments. There are many zinc-binding proteins so the explanation provided by the authors is not convincing. Currently, there is no link between the first (biofilm vs. planktonic growth, 2-DE) and the second part (Nrg1p) of the results. Moreover, only Nrg1p is discussed in Chapter 4 (discussion).

- Why weren't the differences in expression quantified e.g. through ELISA?

4. Discussion

The authors do not discuss all of their findings but concentrate on Nrg1p. What about Spe3, Mdh1, Sod2, Ywp1, Tdh3?

WHOLE MANUSCRIPT:

  • correct the degree symbol;
  • correct multiple grammar errors.

Author Response

Comments and Suggestions for Authors

The presented manuscript of A. Paudyal and G. Vediyappan about the cell surface protein expression in C. auris is interesting to read and deals with an important topic for the better understanding which factors may play a role in C. auris biofilm formation.

Author’s Response: We sincerely thank the reviewer for helpful comments and valuable time. We carefully read all the comments and addressed them in the revised manuscript. Additional details are included in the updated version which hopefully addresses the concerns. The manuscript is thoroughly revised and checked for grammatical errors. All the modifications were indicated with track changes in the attached word file.

We hope the revised version is improved and acceptable to the reviewer.

  1. First (1) they try to analyze the overall cell surface protein expression of planktonic and biofilm cells. Second (2) they focus on Nrg1p expression which seems a known relevant factor (in C. albicans) in biofilm formation. In the introduction, nearly any information about Nrg1p is missing. The significance of Nrg1p in biofilm formation has to be described and documented by references.

Author’s Response: We thank the reviewer for pointing out our oversight of information on Nrg1p in the introduction section. We have included information about Nrg1, including its role in biofilm formation in the introduction (lines 45-55) with appropriate references.

  1. In the results section under the concept of (1) you have found 5 proteins highly expressed in biofilm C. auris cells. The presented experimental data are convincing. But the importance of these 5 proteins in C. auris biofilm formation remains unclear for the reader. Maybe the authors can carry out or at least propose experiments that may validate the real significance of these 5 proteins in C. auris biofilm formation.

Author’s Response: We recognize the reviewer’s point on the role of these 5 proteins that are overexpressed in biofilms. As a first step, we showed the differential expression of cell surface proteins in C. auris biofilm, and further studies, including deletion of respective genes and determining their roles in biofilm growth, will be required. We have included the possible roles of these overexpressed proteins in C. auris biofilm and pathogenesis in the discussion section (lines 386-403) with necessary references. Future experiments to verify the role of identified proteins are also proposed.

  1. The (2) part dealing with Nrg1p is dominant in the manuscript and should be reflected in the title of the manuscript.

Author’s Response: We agree with the reviewer! The title is modified now (Cell surface expression of Nrg1 protein in Candida auris) to reflect the main aspect of Nrg1p in the manuscript.

  1. The discussion of the findings under concept (1) - the five overexpressed proteins - is also nearly inexistant. In Figure 1 notation of (a) and (b) are missing. 

Author’s Response: As mentioned earlier, we have included a paragraph (lines 386-403) on possible roles of the five overexpressed proteins in the discussion. We have also corrected the missing notation of (a) and (b) for Figure 1.

Round 2

Reviewer 1 Report

The manuscript has considerably improved and its much more focused and  clear. Some further improvement of English style and text editing would be desirable. Labelling of some figures can still be improved.